# High-Compression Crash Simulations and Tests of PLA Cubes Fabricated Using Additive Manufacturing FDM with a Scaling Strategy

Andres-Amador Garcia-Granada

Grup d'Enginyeria en Producte Industrial (GEPI), Institut Químic de Sarrià, Universitat Ramon Llull, E08017 Barcelona, Spain; andres.garcia@iqs.url.edu

**Abstract:** Impacts due to drops or crashes between moving vehicles necessitate the search for energy absorption elements to prevent damage to the transported goods or individuals. To ensure safety, a given level of acceptable deceleration is provided. The optimization of deformable parts to absorb impact energy is typically conducted through explicit simulations, where kinetic energy is converted into plastic deformation energy. The introduction of additive manufacturing techniques enables this optimization to be conducted with more efficient shapes, previously unachievable with conventional manufacturing methods. This paper presents an initial approach to validating explicit simulations of impacts against solid cubes of varying sizes and fabrication directions. Such cubes were fabricated using PLA, the most used material, and a desktop printer. All simulations could be conducted using a single material law description, employing solid elements with a controlled time step suitable for industrial applications. With this approach, the simulations were capable of predicting deceleration levels across a broad range of impact configurations for solid cubes.

**Keywords:** crash; explicit; plasticity; FDM; PLA

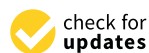



## 1. Introduction

The transportation of goods and people is required to be safeguarded against drops and crashes between moving vehicles. Engineers design countermeasures to ensure that deceleration at various levels does not cause damage to objects or people. Such countermeasures are referred to as crash boxes, crash energy management systems, or deformation elements, as reviewed in [1,2]. The design of these parts is optimized using explicit finite element simulations where kinetic energy is transformed into the deformation energy of the component [3]. This deformation should be plastic, not elastic, to avoid a rebound of the object/person and consequently minimize deceleration and damage/injury values. Hadrys et al. [4] studied deceleration and deformation under real conditions, while Dirgantara et al. [5] numerically and experimentally investigated a crash box structure with holes. These holes decreased the initial peak force prior to buckling, thereby reducing decelerations levels. Most designs are based on shell structures that bend and buckle like tubes or egg boxes. However, some countermeasures are designed using solid parts, such as foam. Lyn et al. [6] conducted headform drop tests against polyurethane crash mats. They noticed in their research that accounting for the test velocity is important to achieve accurate simulation results due to the viscoelastic effect of air escaping from inside the foam. Fragoso et al. [7] explored the use of aluminum foam to enhance crash safety by reducing dash panel intrusions and the acceleration gradient.

New complex designs have become feasible with the introduction of additive manufacturing techniques, also known as 3D printing. Prabhu et al. [8] explored the new possibilities of design using additive manufacturing to enhance creativity. Most engineers are accustomed to design for manufacturing and assembly (DFMA) [9], but we can now consider design for additive manufacturing (DFAM) [10]. This means that many parts

that require the creation of tools and assemblies can be simplified, reducing the number of parts. However, basic knowledge of how these AM parts behave is required. Early approaches for these new opportunities included the study of designing cores for shell tubes mimicking bone structure. Hou et al. [11] conducted crash tests on 3D-printed lattice structures serving as cores of aluminum shells. They analyzed the performance of four different lattice structures to optimize and maximize specific energy absorption and minimize weight. Ingrole et al. [12] investigated bioinspired energy-absorbing material designs using additive manufacturing, defining eight different designs in nature that could be replicated using additive manufacturing capabilities for freedom of design. San et al. [13] performed a review of recent research on bio-inspired structures and materials for energy absorption applications, comparing traditional fabrication methods with additive manufacturing capabilities. Polylactic acid (PLA), a biodegradable thermoplastic made from corn starch, is a widely used material for desktop Fused-Deposition-Modeling (FDM) printers [14]. The use of PLA is popular as this material adheres well to a printing bed at room temperature, yielding robust but relatively brittle parts. PLA is the easiest material to use in this domain, inducing low warping while printing at low temperature. This research tries to provide basic knowledge about the high-compression-energy absorption of PLA cubes that can be easily scaled without requiring a new mold for casting or plastic injection. There are studies on numerical analyses conducted under quasi-static conditions for PLA structures. For example, Bintara et al. [15] examined the deformation pattern and energy absorption of polylactic acid (PLA) carbon crash boxes under quasi-static loading. They used tubes with an inner diameter of 51.04 mm with several wall thicknesses (1.6, 1.8, and 2 mm) and a length 120 mm, applying compression at a rate of 2 mm/min ($0.000277$ s$^{-1}$). The corresponding force–displacement curves displayed a high initial, undesirable peak of force. Chen et al. [16] performed an experimental and numerical investigation of 3D-printed PLA origami tubes printed at 5 mm/min, but, once again, this was conducted under quasi-static uniaxial compression. Quanjin et al. [17] studied the effect of the infill pattern, density, and material type of 3D-printed PLA cubic structures, also under quasi-static loading. They similarly observed an undesirable high peak in force. Yousefi et al. [18] researched additively manufactured, foamed polylactic acid for their potential use as lightweight structures with varying densities by altering deposition temperatures. In their study, they performed tensile and bending tests at a low speed rate of 2 mm/min, demonstrating an improvement in stiffness induced by incorporating sandwich beams with a shell-and-foam core. Strain rates were not reported in this research. Sajadi et al. [19] tested various structures made of PLA, showing a desirable, nearly flat plateau at up to 50% compression and reporting specific energy absorption for each design. Simulations were executed at a very high strain rate of $10^{10}$ s$^{-1}$ using molecular dynamics (MD), while tests were performed at 2 mm/s for a cube of size 50 mm, which led to a low strain rate of $2/50 = 0.04$ s$^{-1}$. Rahman et al. [19] explored the optimization of FDM manufacturing parameters for assessing the compressive behavior of cubic lattice cores using an experimental approach via the Taguchi method. They used a compression speed of 2 mm/min for a cube with a size of 30 mm, which led to a low strain rate of $0.0667$ s$^{-1}$. Searching for porous structures tested at high strain rates, we find research for bones. Real bones tested at high strain rates between $10^{-3}$ and $10^{3}$ s$^{-1}$ by Qiu et al. [20] showed a large influence of strain rate on Young's modulus and ultimate stress compared with fresh bone dehydrated and kept in formalin.

The literature does not provide information pertaining to the compression of cubes of PLA under dynamic loads that can be used to obtain real stress–strain behavior for crash energy management. The aim of this paper is to test and validate simulations of compression to show a real plateau and use this information to simulate a real impact for energy absorption. In this research, we want to achieve several different objectives:

1. Verify the repeatability of compression force–displacement curves for cubes fabricated using additive manufacturing;
2. Verify the scalability of experimental curves to obtain a unique stress–strain curve for all sizes;

3. Verify simulations of compression with a unique stress–strain curve and providing a good correlation with experimental results;
4. Develop a protocol for assessing dimension mass and velocity for impacts on each cube side;
5. Validate explicit simulations with mass scaling to adjust time steps;
6. Compare the use of foam materials with conventional plasticity models;
7. Validate simulations with large strain rates.

## 2. Materials and Methods

This research differentiates between the experimental process of fabricating specimens, testing such specimens in terms of compression, and simulations of compression and the impact on such specimens. Cubes were designed with side lengths of *L* = 10, 15, 20, and 25 mm to obtain a complete set of specimens to allow different levels of energy to be managed.

### 2.1. Fabrication of PLA Cubes

For the fabrication of PLA cubes, a common desktop printer, the Prusa i3mk3 (Prusa3d, Prague, Czech Republic), was used to maximize the reproducibility of this research. Filament was purchased from Amazon Basics PLA (filament diameter: 1.75 mm; roll weighing 1 kg). Slicing was performed using Ultimaker Cura 4.8.0 with a 100% infill with ±45° lines and layers of 0.2 mm. Figure 1 shows odd and even layers of the slicing. An isometric view with slices of all layers is also provided. The colors indicate the outer skin (red), closed contour (green), and 100% infill (yellow). As we fixed the number of outer-skin pieces and closed contours, the percentage of infill was lower for smaller parts. For *L* = 10, only 49% of the cross section is infilled, with just 13 connection points between infill and closed contour. This reduced number of contact points could result in a lower stiffness. For the small cube for which *L* = 10 mm, a total number of 50 layers were required (10/0.2), while for a large *L* = 25 mm cube, a total number of 125 layers were necessary (25/0.2). For each cube size, a total of 20 specimens were fabricated. The time and grams required to fabricate each cube, according to Ultimaker Cura, are presented in Table 1.

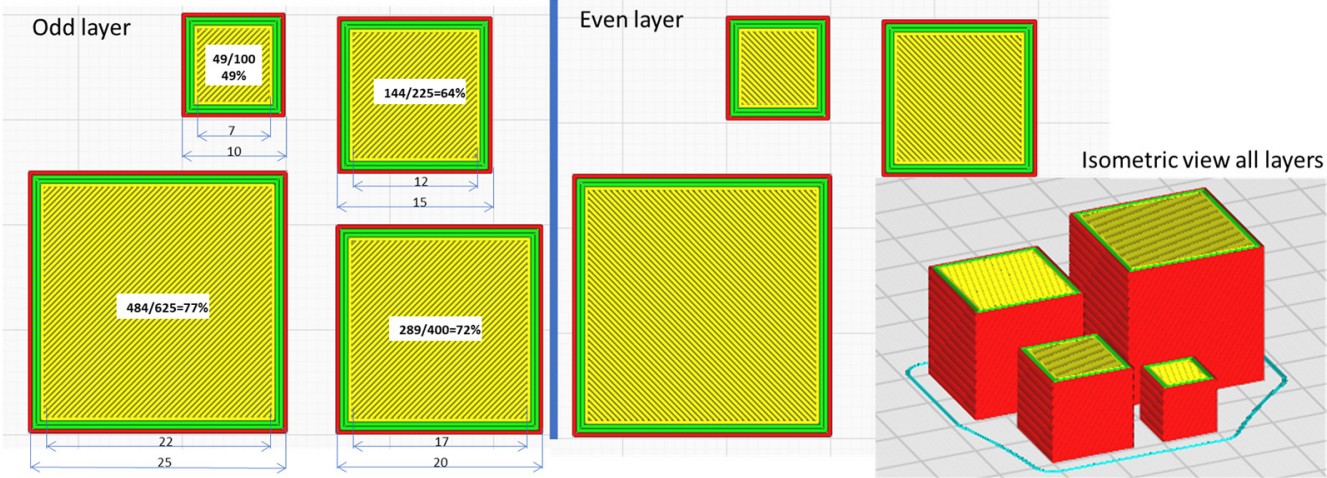

**Figure 1.** Ultimaker Cura slicing for odd and even layers and an isometric view of all layers.

**Table 1.** Cube size with scale factor ratio "*s*" compared to *L* = 10 mm in brackets, fabrication time, theoretical weight, and actual weight.

| *L* [*s*] (mm) | Time (min) | Theoretical Weight (g) | Actual Weight (g) | Actual Density (kg/m³) |
|---|---|---|---|---|
| 10 [×1.0] | 8 [×1.0] | 1 | 1.241 ± 0.018 or ±1.45% | 1240 |
| 15 [×1.5] | 25 [×3.1] | 4 | 4.100 ± 0.008 or ±0.20% | 1215 |
| 20 [×2.0] | 59 [×7.4] | 10 | 9.660 ± 0.036 or ±0.38% | 1207 |
| 25 [×2.5] | 114 [×14.2] | 19 | 18.820 ± 0.082 or ±0.44% | 1204 |

Real statistics for weights of parts are given in Table 1. For the column concerning cube size, in brackets, we have the scaling factor *s*. For fabrication time, it is expected that this time would be proportional to the volume ($s^3$), as is anticipated for weight.

*2.2. Compression Tests of PLA Cubes*

Compression tests were conducted between parallel plates using an Instron 5985 machine (Instron, Norwood, MA USA) with a load cell capable of recording up to 50 kN. Displacement control was implemented at strain rate of 0.02 s⁻¹ (ranging from 0.2 mm/s for *L* = 10 mm up to 0.5 mm/s for *L* = 25). For all tests, displacement, force, and time were recorded every 0.01 s (100 Hz). Tests were performed at up to 80% compression once the force plateau was reached and all specimens showed a large, undesirable load increase. Therefore, the time required to test the sample was 0.8/0.02 = 40 s, yielding 4000 data points.

Samples were labeled with a black arrow, indicating the additive manufacturing layer fabrication direction, and a red arrow, indicating the loading direction. Photographs were taken during the test to analyze the deformation based on orientation and the cross-section increase during compression.

Figure 2 depicts the setup of compression test at the initial stage, once layer separation was evident, and a comparison of loading direction and two different cube sizes after load removal.

Initial compression

Layer separation for perpendicular

Size and loading direction effects

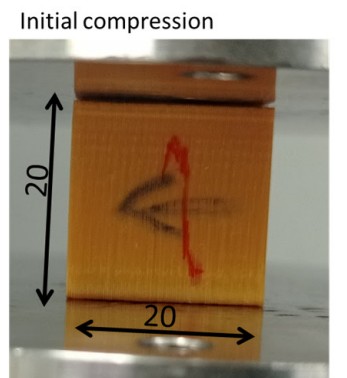
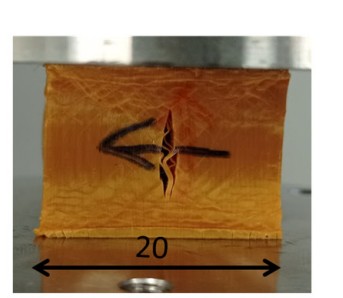
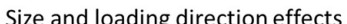
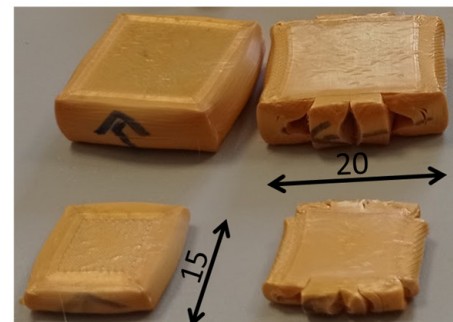

**Figure 2.** Compression testing of cubes of different sizes in different directions for *L* = 15 and 20 mm.

A datasheet was used to convert force into stress and displacement into strain for each specimen. Measurements of deformed parts were taken to estimate Poisson's ratio. The equation for the elastic isotropic matrix is as follows:

$$\left\{ \begin{matrix} \varepsilon_x \\ \varepsilon_y \\ \varepsilon_z \end{matrix} \right\} = \frac{1}{E} \begin{vmatrix} 1 & -\nu & -\nu \\ -\nu & 1 & -\nu \\ -\nu & -\nu & 1 \end{vmatrix} \left\{ \begin{matrix} \sigma_x \\ \sigma_y \\ \sigma_z \end{matrix} \right\} \tag{1}$$

where $\varepsilon$ is deformation, $E$ is Young's modulus, $\nu$ is Poisson's ratio, and $\sigma$ is stress. When we only have stress in the $z$ direction, then we can solve for the elastic region

$$\varepsilon_x = \varepsilon_y = \frac{-\nu}{E}\sigma_z = -\nu\varepsilon_z \Rightarrow \nu = -\frac{\varepsilon_x}{\varepsilon_z} \tag{2}$$

and the volume change can be estimated as

$$Vc = (1+\varepsilon_x)\left(1+\varepsilon_y\right)(1+\varepsilon_z) - 1 = (1-2\nu)\varepsilon_z + \left(\nu^2 - 2\nu\right)\varepsilon_z^2 + \nu^2\varepsilon_z^3 \tag{3}$$

Which means that for $\nu = 0$, the volumetric change is proportional to the imposed deformation, and when $\nu = 0.5$, the volumetric change depends on terms that are small for small deformations of $-0.75\varepsilon_z^2 + 0.25\varepsilon_z^3$. For $\varepsilon_z = 0.1$, the volumetric change is just $-0.00725$, but for $\varepsilon_z = 0.5$, the volumetric change is mathematically significant, $-0.156$ ($-15.6\%$).

Rubber materials are considered almost incompressible as $\nu$ is close to 0.5 with values of around 0.4999, while foam is considered to have limited lateral expansion with Poisson's ratio values around 0.1. Conversely, Grasso et al. [21] demonstrated that PLA exhibits a Poisson's ratio of 0.35 for pure filaments.

However, the most important region for energy absorption is created once the material goes beyond yield stress with increasing plastic deformation. Most material models use an assumption of incompressibility (similar to $\nu = 0.5$ for an elastic model and small strain) during plasticity, while other specific models, such as those used for foam, assume there is no change in lateral expansion during plasticity (similar to $\nu = 0$ for an elastic model). This has a clear implication in the translation from force and displacement into stress and strain. For incompressibility, when $\varepsilon_z = -0.8$ (80% compression), $\varepsilon_x = \varepsilon_y = 0.4$, and therefore the area is increased by $(1 + 0.4)^2 = 1.96$, and for the same constant (plateau) force, stress should be reduced by $1/1.96 = 0.51$ at 80% compression. As mentioned above, this introduces a large error if the large degree of strain is not calculated accurately.

In the case of lateral expansion, friction is important, as in this case the loading is not completely uniaxial in compression. Friction was measured between cubes and steel plates using an inclinable table and by measuring the static and dynamic friction coefficients. Simulations with values of 0.01 and a static friction value of 0.3 were carried out, obtaining the same results due to the fact that main forces are applied in the normal–vertical direction. The setting of friction equal to 0 was not used to obtain stability of the cube, preventing lateral movements without requiring additional boundary conditions.

The first objective was to check the repeatability in terms of force–displacement for each geometry and orientation. The second objective was to verify if stress–strain curves were very similar despite the size and orientation of specimens. The third objective was to check if simulations with a unique stress–strain curve would provide an acceptable correlation.

The fourth objective was to determine the required mass and velocity for impact tests for each scale factor ratio "$s$". It should be noted that area is scaled by $s^2$ and mass and volume by $s^3$. Using the datasheet for force and displacement, energy was monitored for each specimen up to around 50% deformation to avoid impacts entering the densification zone where a steep slope of force displacement is reached. The expected scaling for energy should be proportional to scaling force and displacement and therefore scaled by $s^3$. This means that it was necessary to check if we obtained the same energy/mass ratio for all geometries and orientations. Once we selected the amount of energy for impact, we chose the impact mass and velocity. For small specimens, we fixed the average acceleration to be in the range of measurable 50 g and calculated the minimum mass. After the examination of results, it was found that for small specimens, we obtained a mass of 16 kg and velocity of 1.8 m/s, while for a large specimen, these values were 40 kg and 4.5 m/s. For other geometries, mass and velocity were scaled by ratio $s$, and therefore energy was scaled by ratio $s^3$. For the maximum ratio $s = 2.5$, we expected to obtain $2.5 \times 50$ g $= 125$ g, which is acceptable for accelerometers before the detailed examination of experiments. Impact

time was also estimated as $t = v/a$ assuming constant deceleration. With this, we expected to have impacts of $1.8/(50 \times 9.805)$, approximately 3.6 ms. All these initial numbers are fundamental to acquire accelerometers and signal-monitoring systems of high frequency.

Using the real force and displacement curves from tensile tests, a theoretical impact acceleration was estimated as $F/m$ for each of the 4000 data points, and, using integration within the data sheet, it was possible to obtain velocity $v$, displacement $d$, and time $t$, serving as a closer estimation for future impact tests.

### 2.3. Simulation of Compression and Impact Tests of PLA Cubes

Simulations were conducted using the explicit software ESI® Virtual-Performance (previously known as PamCrash) version 2019 software run on an HP Envy Laptop with a 4-core Intel® Core™ i5-10300H CPU @2.5 GHz. An explicit solution where nodes are constantly updated was used, thus yielding real cross sections and lengths. Therefore, the software uses the true stress and true strain approach.

Coherent units were used for length (mm), for mass (kg) and for time (ms). Therefore, we derived other units such as velocity (mm/ms) = (m/s), acceleration (mm/ms$^2$), stress (GPa), force (kN), energy (J) and density (kg/mm$^3$).

The compression test was simulated using 20 ms with a force–displacement curve that reached 80% at 10 ms and then returned to the original position. With a forced time step of 0.1 µs to complete the 20 ms simulation, it was necessary to compute $20{,}000/0.1 = 200{,}000$ time steps, while a forced time step of 5 µs required only 4000 time steps (i.e., 50 times less). Time history was recorded every 5 µs to obtain a total of $20{,}000/5 = 4000$ data points. For $L = 10$ mm, the speed was 800 mm/s, while for $L = 25$ mm, the speed was therefore 2000 mm/s. The strain rate for all compression tests was 80 s$^{-1}$.

The cube was simulated with 125 ($5 \times 5 \times 5$) perfect solid cube elements with $6 \times 6 \times 6 = 216$ nodes each. Top and bottom plates were simulated using a large shell element connected to a rigid body. This step was executed to ensure that the material used in the compression tests was not deformed compared to our PLA cubes. The center of gravity for each rigid body was constrained as a boundary condition to avoid any displacement or rotation. Only the $z$ displacement (3rd direction) was set to be free for the moving plate.

Scaling of the mesh with ratio $s$ was implemented to compute all cube sizes with the same forced time step. Cubes for which $L = 10$ mm used elements $10/5 = 2$ mm in size, while cubes of $r = 2.5$ and $L = 25$ mm used elements of $25/5 = 5$ mm in size after the applied scaling.

Natural time step for the adjusted PLA density and stiffness was estimated in 1 µs for a 2 mm element length for small cubes and 2.5 µs for cubes with a 5 mm element length for large cubes. However, if during impact the element was compressed by 90%, then the time step could be reduced from 1 to 0.1 µs as mesh length would be reduced from 2 to 0.2 mm. The fifth objective was to validate the time step control strategies. Three different strategies were used for time step control.

- The first time step was forced to slow down to 0.1 µs, which is the worst-case scenario for small cubes compressed by 90%.
- The second approach utilized a free time step, which changed during compression.
- The third approach involved mass scaling to force the time step to be 5 µs. This entails that if a maximum compression requires a time step of 0.1 µs, density must be multiplied by a factor of $(5/0.1)^2 = 2500$. This is an extreme case of mass scaling where the mass of 1 g of small cubes will result in 2.5 kg, which could derive non-negligible differences in force.

The equation for stable time step increment is

$$\Delta t = kL\sqrt{\frac{\rho}{E}} \tag{4}$$

where $k$ is calculated as the function of damping ratio, $L$ is the minimum mesh element length, $\rho$ is material density, and $E$ is elastic stiffness expressed as Young's modulus. For the damping, we assume

$$k = \sqrt{1 + \xi^2} - \xi \tag{5}$$

where $\xi$ is the damping ratio, usually defined as 0.1, used to achieve simulation stability and avoid large resonance magnification of deformations. However, Perez-Pena et al. [22] provided a simple procedure for obtaining real damping values with which to feed simulations, showing that stiff steel materials exhibited low damping of around 0.001 and that the damping values of polymer materials, similar to PLA, were in the range of 0.01. Simulations were carried out assuming this low damping value.

Magnification factor of vibrations in resonance is as follows:

$$M = \frac{1}{2\xi} \tag{6}$$

where $\xi$ is the damping ratio. For a damping ratio of $\xi$ = 0.01, we can magnify vibrations by $M$ = 50, which could produce large instabilities during the simulation.

Solid elements with two different material properties were used. The sixth objective of this research was to compare the use of foam materials with common elasto-plastic approaches. First, the material was assumed to behave like foam. In this case, material type 45 was selected, defining the stress–strain curve directly from the experimental force and displacement curve. The second approach considered PLA to be incompressible; therefore, a common elastic-plastic material was used to introduce stress versus plastic strain. Experimental force–displacement curves were used to consider a scenario where the cross-sectional area increased proportionally to compression to maintain constant volume. The main objective of using both approaches was to examine the benefits and risks of each approach.

Material type 45 requires the definition of Young's modulus $E$ and assumes $\nu$ = 0, while material type 1 requires the use of shear modulus $G$ and bulk modulus $K$. The relationship between both parameters is as follows:

$$K = \frac{E}{3(1 - 2\nu)}; G = \frac{E}{2(1 + \nu)}; E = \frac{9KG}{3K + G}; \nu = \frac{2K - 2G}{2(3K + G)}; \tag{7}$$

Using solid elements introduces the risk of obtaining an undesirable error of negative volume. This occurs when the positions of upper nodes calculated for a time step are below the lower nodes. As stated in the design scenario, all impact velocities are proportional to scaling ratio $s$. The primary challenge is designing for compression carried out at 1.8 mm/ms. This means that for a time step of 5 μs, the nodes travel 0.009 mm, and the software should be able to avoid negative volume problems even for a compressed mesh with a size of 0.2 mm.

A Python script was developed to scale the simulations on input decks, run the simulations using a batch command, and post-process all simulations using a session file to generate pictures of contour-deformed mesh, energy curves, force–displacement curves, and acceleration curves, which were then saved as ascii files.

Figure 3 shows the model used for compression and impact tests. The top node is free to move in $z$ direction. The bottom node is fixed in all directions. These nodes are used as the center of the rigid body connecting to the top and bottom plates, respectively. For the isometric view, transparency was introduced for the top and bottom plates.

In the simulations for impact tests, we aimed to achieve the seventh objective, namely, the validation of curves for high strain rates.

The mass and velocities with respect to dimensions for each experimental test and simulation impact are provided in Table 2 as a result of the energies from the compression experiments. This procedure corresponds to the fourth objective of this research, as all

parameters can be estimated from the scale factor "$s$". Experiments were carried out using a drop height estimated from the desired impact velocity.

**Table 2.** Procedure for impact energy management.

| Input | $m$ (kg) | 16 | 24 [$\times 1.5$] | 32 [$\times 2.0$] | 40 [$\times 2.5$] | $16 \times s$ |
|---|---|---|---|---|---|---|
| Input | $v$ (m/s) | 1.8 | 2.7 ($\times 1.5$) | 3.6 [$\times 2.0$] | 4.5 [$\times 2.5$] | $1.8 \times s$ |
| $mv^2/2$ | $E$ (J) | 25.92 | 87.48 [$\times 1.5^3$] | 207.3 [$\times 2.0^3$] | 405 [$\times 2.5^3$] | $25.92 \times s^3$ |
| $10(E/25.92)^{1/3}$ | $L$ (mm) | 10 | 15 [$\times 1.5$] | 20 [$\times 2.0$] | 25 [$\times 2.5$] | $10 \times s$ |
| $v^2/(2g)$ | $h$ (mm) | 165.5 | 372.8 [$\times 1.5^2$] | 662 [$\times 2.0^2$] | 1034 [$\times 2.5^2$] | $165.5 \times s^2$ |

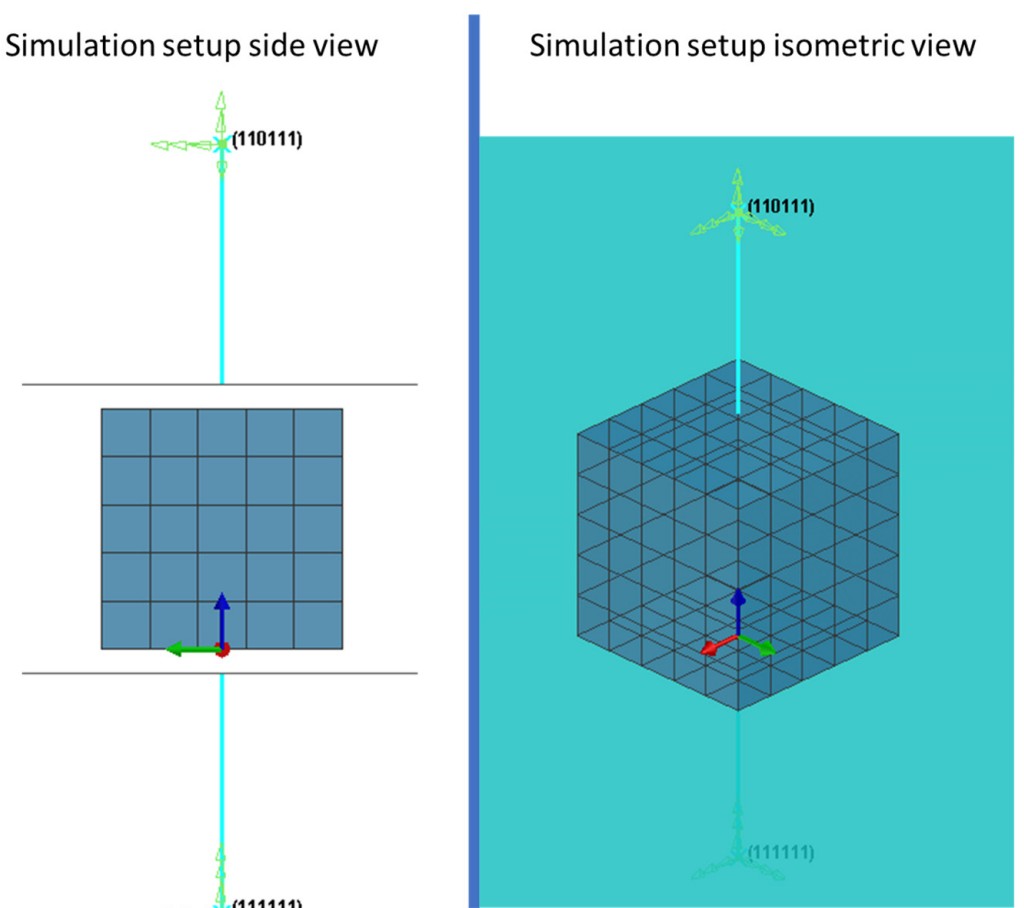

**Figure 3.** Simulation of compression tests of cubes of different sizes in different directions.

## 3. Results

The first objective was to verify the repeatability of all the compression tests in order to obtain the values of forces with a normal distribution. The samples with more dispersion in the results were the smallest cubes with a size of $L = 10$ mm, for which tolerances have a larger influence. In Figure 4, up to five different tests are shown, exhibiting very similar behavior. The standard deviation for forces was calculated, yielding in the worst case $\pm 3.6\%$.

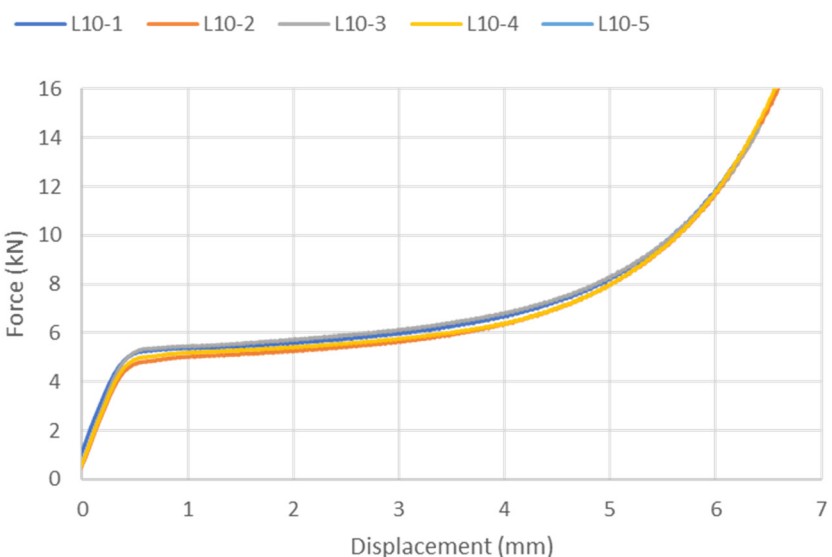

**Figure 4.** Repeatability of experimental compression tests for *L* = 10 mm.

The second objective was to check if the orientation of loading according to fabrication was important and if we could scale our specimens to use the same stress–strain curve. Figure 5 shows four tests concerning fabrication direction and four tests concerning a perpendicular direction (named Test_10p, Test_15p, Test_20p, and Test_25p). All the force–displacement curves were converted to stress–strain curves for comparison. Only the smallest specimens for which *L* = 10 mm in the fabrication direction showed a different behavior. All the other specimens were very similar. The biggest specimen, with a size of *L* = 25 mm (A = 625 mm$^2$), showed a plateau at around 80 MPa. This is because 80 × 625 = 50,000 N = 50 kN was the maximum force allowed by the tensile test machine.

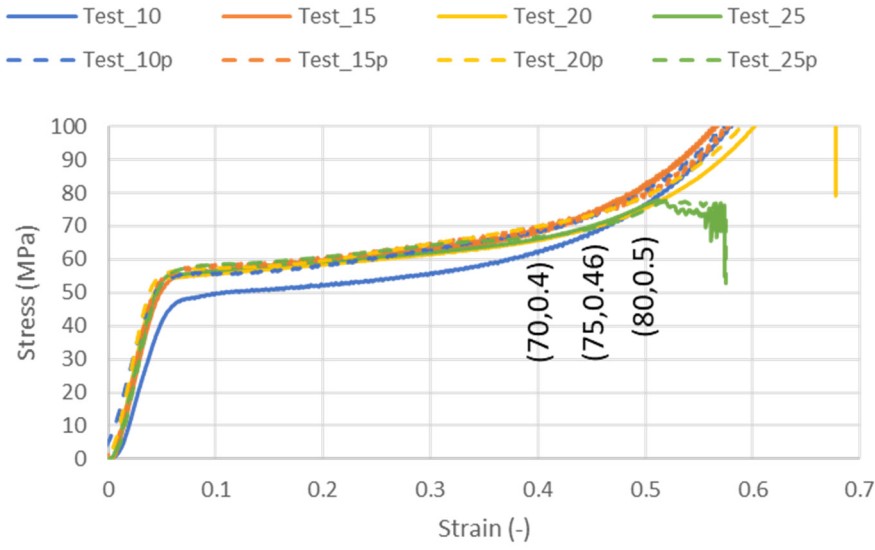

**Figure 5.** Effect of orientation in experimental compression tests for all sizes.

Figure 5 shows that the stress for 40% compression was around 70 MPa; for 46% we reached 75 MPa, and for 50%, we started to leave the plateau region, increasing stress to up to 80 MPa.

Referring to the experimental stress–strain curve, we fed the simulation model using material type 45, which is used for foams. It was also simulated as a common type 1 elastic plastic solid with a stress-versus-plastic-strain curve. The elastic part is not very important for energy absorption, but *E* was estimated to be 1.75 GPa for material type 45 with *ν* = 0.35

during the presentation of elastic behavior, leading to $K$ = 1.944 GPa and $G$ = 0.6481 GPa for material type 1. The values used in both simulations are summarized in Table 3.

**Table 3.** Stress–strain curve for material type 45 and type 1 in PamCrash.

| Foam 45 | Strain (-) | 0 | 0.032 | 0.4 | 0.5 | 0.6 | 0.7 | 0.8 | 0.9 | 0.99 |
|---------|-----------|---|-------|-----|-----|-----|-----|-----|-----|------|
|         | Stress (GPa) | 0 | 0.056 | 0.07 | 0.08 | 0.1 | 0.14 | 0.22 | 0.38 | 2 |
| Plastic 1 | Pl.Strain (-) | | 0 | 0.375 | 0.476 | 0.576 | 0.675 | 0.774 | 0.878 | 0.978 |
|           | Stress (GPa) | | 0.056 | 0.042 | 0.043 | 0.044 | 0.051 | 0.058 | 0.071 | 0.091 |

The integration of force displacement for all the tests allowed us to create a unique energy density curve for compression strain as shown in Figure 6. For 40% compression, it was found that PLA cubes could withstand 18 kJ/kg or 21.6 J/cm³ (1 cm³ = volume of basic cube with $L$ = 10 mm = 1 cm). For 46% compression, it was found that PLA cubes could withstand 21.6 kJ/kg or 25.92 J/cm³. For 50% compression, it was found that PLA cubes could withstand 25 kJ/kg or 30 J/cm³.

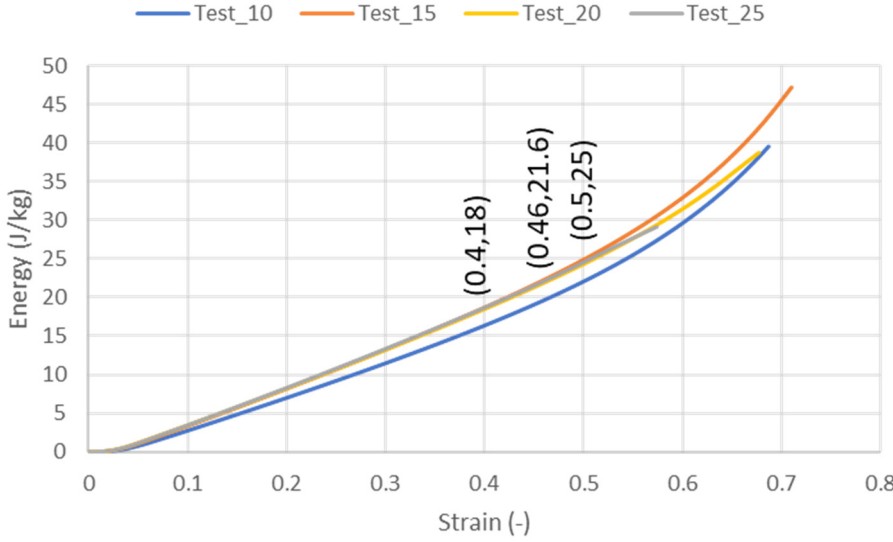

**Figure 6.** Energy density obtained from experimental compression tests for $L$ = 10, 15, 20, and 25 mm. Energy values at 40%, 46%, and 50% are provided for impact estimation.

The third objective was to compare the experiments with the simulation model using just one stress–strain curve. Figure 7 shows the force–displacement curves for the experiments (Test) and compares them with those from the simulations (SIM) of this compression test for material type 45 and a time step of 0.1 μs. A good correlation was achieved during the plateau phase, while differences were larger when the load increased in the densification area. It should be noted that the main objective was to obtain a similar force for the plateau area, where maximum accelerations are expected during an impact if the cube is able to absorb the energy before reaching the steep slope of force–displacement. Note that the simulations included the unloading path, but there was good agreement for all the compression paths within the plateau with the results of the experiments.

These differences in force and displacement can be minimized in a stress–strain plot. Figure 8 shows that all the specimens are in the same stress region, but it helps to understand that the agreement between the results of the simulation and experiment is acceptable for the region of interest for the impact tests wherein we aimed to obtain a close to constant force (plateau). All simulations have different force–displacement characteristics, as shown in Figure 7, but they overlap in force 8 when we converted to stress and strain.

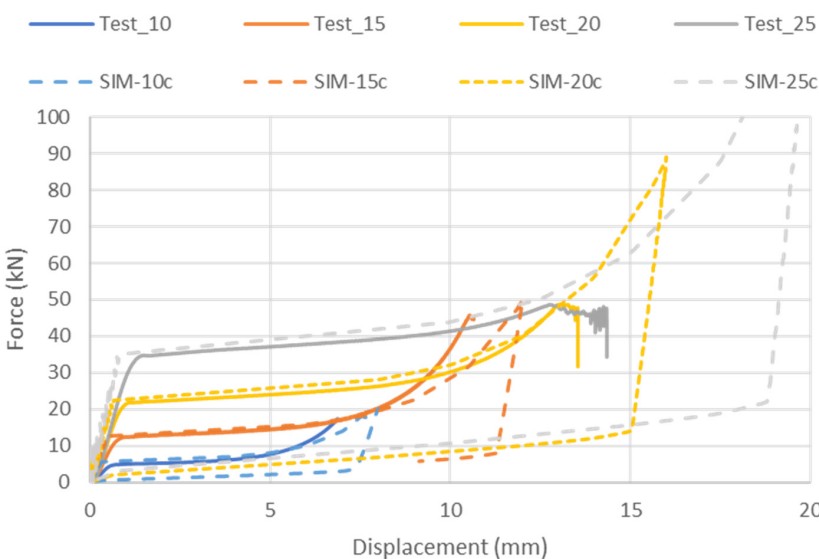

**Figure 7.** Force-versus-displacement curves for experimental tests and compression simulations for *L* = 10, 15, 20, and 25 mm.

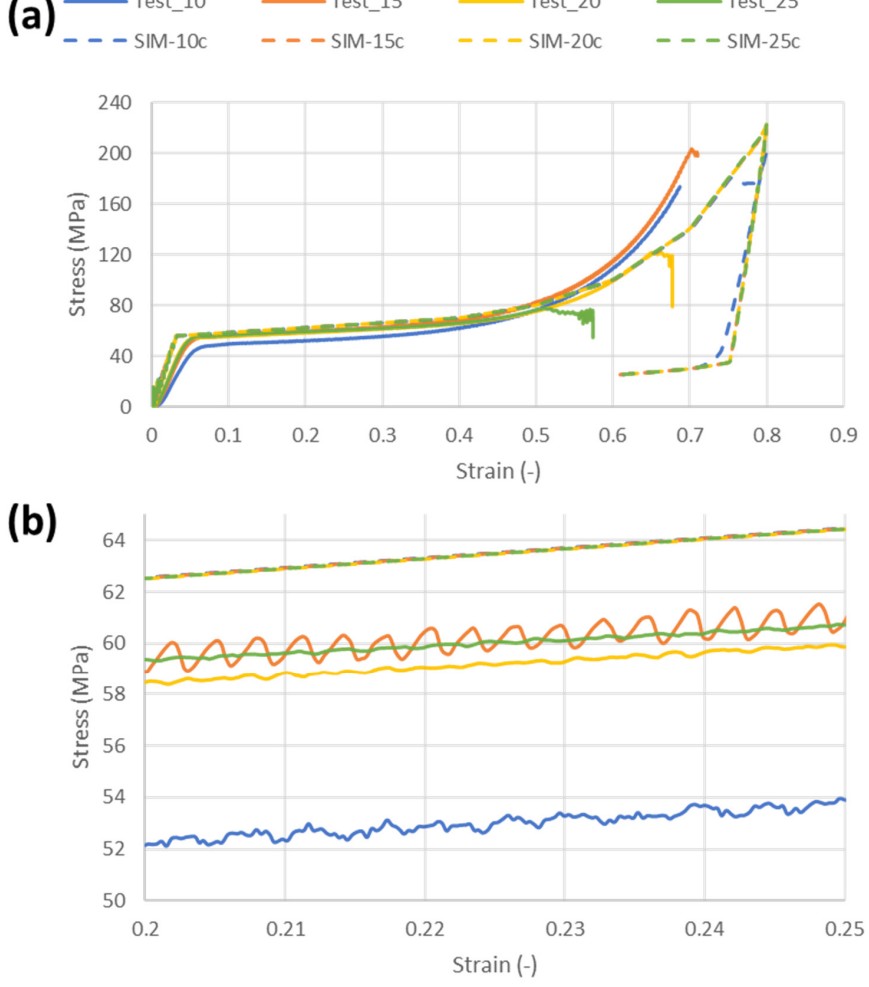

**Figure 8.** Stress–strain curve from experiments and compression simulation: (**a**) full scale and (**b**) detail for 20–25% strain.

Referring to the details of the stress–strain curve for 20–25% strain shown in Figure 8b, we can see the error in the simulation by using a unique stress–strain curve for all the components. For $L = 10$, the error when comparing the experiment and simulation should be around $(62.5 − 52)/52 = 20\%$. For $L = 15, 20$, and 25, the error should be around $(62.5 − 59)/59 = 6\%$. Obviously, this error corresponding to the use of a unique stress–strain curve for simulation could be computed for other values of deformation, but the main objective was to obtain a good correlation for impacts. For large compression values, the simulation provides smaller values of stress for $L = 10$ and 15. The $L = 20$ and 25 specimens could not achieve large compression values due to limitations in experimental maximum force. The lack of agreement from 60% compression could be attributed to the initiation of cracks on the specimen and changes in the cross section that were not taken into account when calculating stress from experimental force.

The fourth objective was to demonstrate that with average values of acceleration, we can estimate impacts, but we require the complete integration of experimental force–displacement curves to obtain a real estimation of acceleration and time during impact. Figure 9 shows the acceleration curves estimated from the experimental force–displacement curves compared with those obtained from simulations. There is an agreement in the estimation of the maximum acceleration level, with the simulations slightly overestimating these values. It should be noted that the compression experiments did not consider the unloading path; therefore, the experimental estimation is only able to predict acceleration levels up to the maximum value of acceleration, when unloading should start.

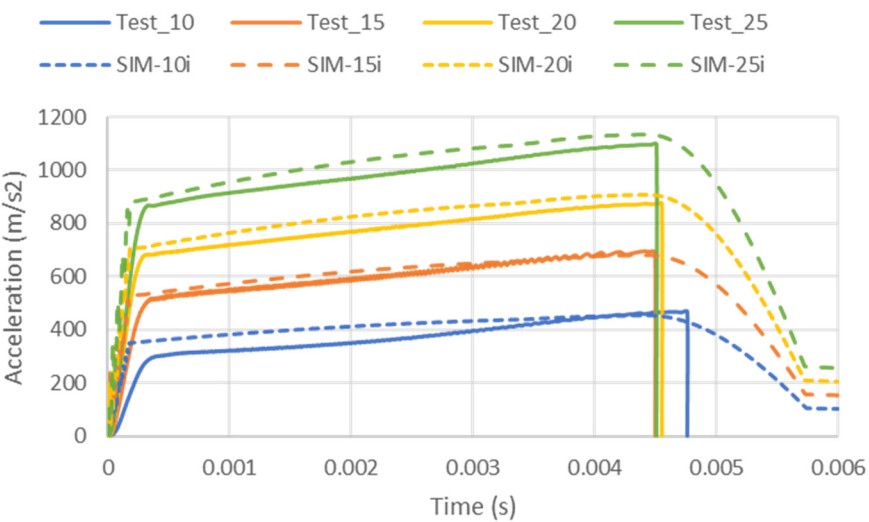

**Figure 9.** Acceleration–time curves obtained from the integration of experiments (Test) and impact simulations (SIM).

The fifth objective was to validate the time step strategy. Table 4 shows the mass increase when forcing the time step to go faster than a stable time step. This mass increase for the PLA cubes seems relevant for compression tests, but if we consider that a cube weighing 1 g could be mass-scaled to reach 90.2 g (9020%), this increase is negligible when impacting a mass of 16 kg (0.56%). Note that this mass increase could be larger for higher compression when elements become smaller. Table 4 shows the advantage of using mass scaling to implement a larger time step by reducing CPU computational time for an impact from 19 to 6 s (around one third of the time). This is very important to obtain results in a reasonable amount of time.

**Table 4.** Mass increase and CPU time for each simulation as a function of time step.

| | Compression | | | Impact | | |
| --- | --- | --- | --- | --- | --- | --- |
| | Ts = 0.1 µs | Ts = free | Ts = 5 µs | Ts = 0.1 µs | Ts = free | Ts = 5 µs |
| Mass increase compression (%) | 0% | 0% | 9020% | 0% | 0% | 6280% |
| Mass increase impact (%) | 0% | 0% | 0.56% | 0% | 0% | 0.38% |
| N.Time Steps | 200,001 | 38,657 | 4001 | 200,001 | 26,621 | 4001 |
| CPU (s) | 127 | 26 | 8 | 118 | 19 | 6 |
| CPU vs. free (%) | 488% | 100% | 31% | 621% | 100% | 32% |

To validate the time step approach, we plotted the acceleration results for impacts in Figure 10. All the time steps predicted a similar value of maximum acceleration.

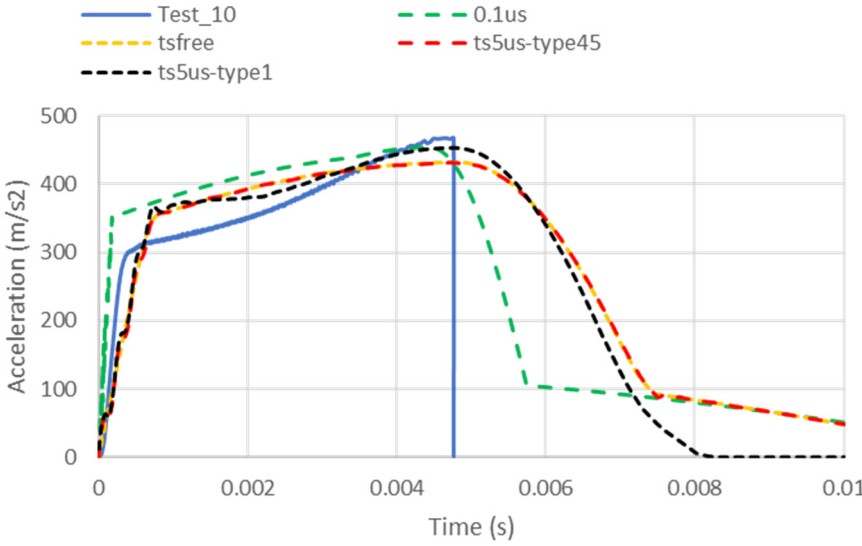

**Figure 10.** Acceleration–time curves from the experiment (Test) compared to those from the simulations using different time steps for the worst correlation case for the smallest *L*, namely, *L* = 10 mm.

Table 5 provides a comparison of the estimated accelerations (calculated as $75L^2/(mg) = 47.8s$) with the values from the experimental test and simulation. Errors in the estimation of acceleration from the simulations compared to those from the experiments are also provided.

**Table 5.** Maximum acceleration results for impact tests and simulations.

| Input | $m$ (kg) | 16 | 24 ($\times 1.5$) | 32 ($\times 2.0$) | 40 ($\times 2.5$) | $16 \times s$ |
| --- | --- | --- | --- | --- | --- | --- |
| Input | $v$ (m/s) | 1.8 | 2.7 ($\times 1.5$) | 3.6 ($\times 2.0$) | 4.5 ($\times 2.5$) | $1.8 \times s$ |
| $mv^2/2$ | $E$ (J) | 25.92 | 87.48 ($\times 1.5^3$) | 207.3 ($\times 2.0^3$) | 405 ($\times 2.5^3$) | $25.92 \times s^3$ |
| $10(E/25.92)^{1/3}$ | $L$ (mm) | 10 | 15 | 20 | 25 | $10 \times s$ |
| $v^2/(2g)$ | $h$ (mm) | 165.5 | 372.8 ($\times 1.5^2$) | 662 ($\times 2.0^2$) | 1034 ($\times 2.5^2$) | $165.5 \times s^2$ |
| $75L^2/(mg)$ | $a$ (g) | 47.8 | 71.7 ($\times 1.5$) | 95.6 ($\times 2.0$) | 119.5 ($\times 2.5$) | $47.8 \times s$ |
| Test | $a$ (g) | 46 | 69 | 86 | 110 | |
| Simulation | $a$ (g) | 45 | 69 | 90 | 115 | |
| (sim-test)/test | % | $-2.2\%$ | 0% | $+4.7\%$ | $+4.5\%$ | |

The sixth objective was to validate material models for type 45 foam and conventional elastic-plastic type1 foam. Both models are shown in Figure 10 for the fastest time step, yielding similar results, wherein acceleration is in agreement with the results from the experimental test.

Figure 11 shows the evolution of the stable time step for the simulations where we used a free time step approach. The time step decreases from about 0.9 µs to values of 0.3 µs in compression and 0.5 µs for impact. This is due to the decrease in the length of hexahedral elements which is inversely proportional to a stable time step.

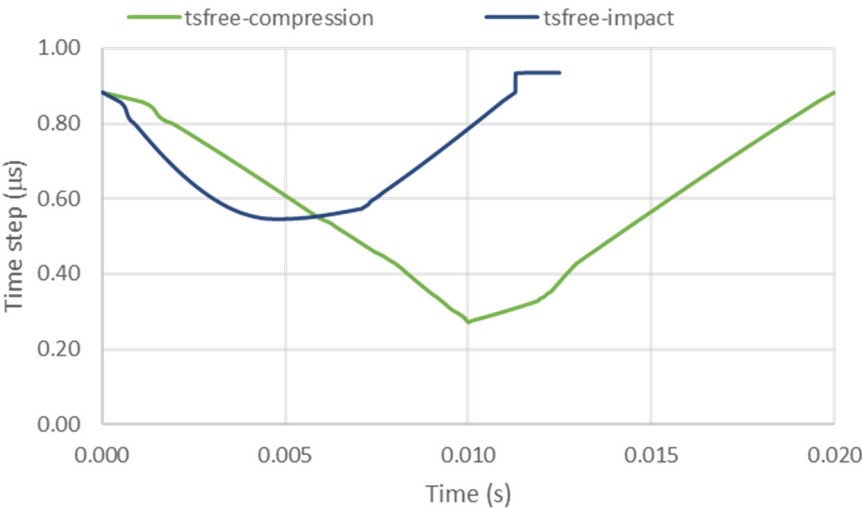

**Figure 11.** Time step used in simulation as a function of simulation time for compression and impact simulation.

The seventh objective was to validate this methodology for large strain rates. In impacts, the cube was compressed to around 5 mm (50% = 0.5 engineering strain) in about 5 ms = 0.005 s, which is an average strain rate of $100\ \mathrm{s}^{-1}$. However, impact tests are not carried out at a constant strain rate. Simulations of this impact allow for the plotting of the engineering strain rate and the true strain rate. The engineering strain rate assumes that the final compression from 4.9 to 5 mm implies a 0.1/10 = 0.01 strain, as the original length of the cube is used. True strain considers the part to be of a real length, calculated as 0.1/5 = 0.02 true strain. Therefore, in large compression, true strain is larger than engineering strain. To calculate explicit finite elements, their updated geometry is used after each time step, providing a true strain rate. Figure 12 shows the strain rate during an impact with a cube for $L$ = 10 mm, m = 16 kg, and v = 1.8 m/s, resulting in 50 J of energy being absorbed.

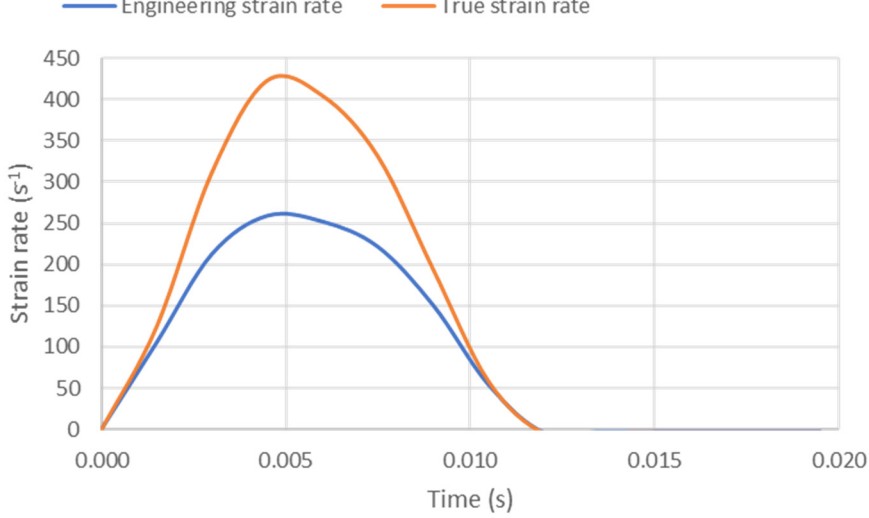

**Figure 12.** Engineering strain rate and true strain rate during simulations of impact on PLA specimen.

With these impacts, we validated the occurrence of high strain rates, which could reach 440 s$^{-1}$, while compression tests were carried out at a low speed of 0.02 s$^{-1}$.

## 4. Discussion

PLA cubes can be used as crash energy management components. These parts can be fabricated via plastic injection. The advantage of additive manufacturing is that it allows us to eliminate stocks, and we can attain the desired cube for a given level of energy with a quicker response and with a lower price for a number of parts below the break-even point. Table 5 provides a summary of all the tests carried out to define how impact energy is transformed into deformation energy. The procedure of scaling by factor $s$ was introduced for each variable slated to be measured. Mass and velocity were scaled by factor $s$, and the energy was calculated, yielding scale factor $s^3$. Using the findings of this research, the required length size of the cube was estimated to be in the region of 46% deformation, which corresponds to 75 MPa, prior to a steep force increase. This length is also proportional to $s$. From an experimental point of view, it is possible to calculate drop height, which is proportional to $s^2$. Finally, it is possible to estimate from the scale factor the expected maximum level of acceleration. This estimation was determined to be $75L^2/(mg) = 47.8s$. These values were compared with the maximum values of acceleration from the tests and simulations, yielding a good agreement.

However, this approach is based on using only 46% of the deformation of the cube. If we try to repeat the same procedure when the cube is compressed into the steep area of the stress–strain curve, then small changes in deformation would imply very large differences in acceleration.

Throughout this paper, it was demonstrated that additive manufacturing could produce repeatable force–displacement curves for large compression rates, including for filament detachment. The sample that clearly differed was the smallest sample measuring $L = 10$ mm, which provided approximately 12% lower stress for the area of the plateau. These small samples provided a compact area of 49 over 100 mm$^2$ (49%), while for $L = 15$, we have a compact area of 144 over 225 mm$^2$ (64%).

The samples can be scaled, yielding a complete set of implications for cross section, volume, mass, energy, etc. These implications are summarized in Table 5, but they require a good comparison of stress–strain curves, as shown in Figure 5. It is also possible to see that orientation has little effect on the stress–strain curves.

The simulations compared well to the compression experiments, as shown in Figure 7 for force–displacement and Figure 8 for the stress–strain curve. The initial elastic slope is quite difficult to measure experimentally because of irregularities in the contact face. The use of preload was avoided, as the desired outcome of the experiment was a complete force–displacement curve for integration to determine absorbed energy. The target of simulations is to provide a good acceleration level during an impact. The correlation seems very promising for the plateau area as well as for the steep slope at up to 50% compression.

For the impact tests, the acceleration-related theoretical values, experimental tests, and simulations compared well, as shown in Figure 9, which was the purpose of this research. The experimental tests did not provide results regarding acceleration during unloading, and it is impossible to compare them once maximum acceleration has been reached. Experimental setups should be improved in the future if one intends to monitor *a3ms* (acceleration hold for at least 3 ms) or *HIC* (head injury criteria) injury values, which require the complete curve of acceleration as in the simulation.

Time step control provided simulations with accurate estimations of acceleration with CPU time reduced to around 31%, as shown in Figure 10. This is a critical point in explicit simulations as a compressed element reduces its length and therefore reduces the time step used for all the simulations. The evolution of the free time step is shown in Figure 11. Fortunately, the good agreement between theory and experiments allowed a good dimensioning of the size of the cube sample to be used. If by mistake a smaller sample size was chosen, then more compression would be induced, and the time step would

decrease to a point where we could not attain results in a reasonable time. Superimposing a time step by means of mass scaling is risky. In this case, we increased the mass of the cube by a factor of 90, from 1 to 90 g. For our impact tests, this was almost negligible, as we applied an impact of 16 kg and therefore obtained good agreement in acceleration.

Also, Figure 10 shows the good correlation of the common elastic-plastic material type 1 and foam material type 45. Foam material is very easy to create as only the unique experimental stress–strain curve from the compression tests is needed. Material type 1 is more difficult to introduce correctly as it requires the creation of a table of stress versus plastic strain, considering large deformations and changes in cross-sectional area during compression.

Finally, Figure 12 shows the strain rate during an impact, which can reach values of up to 440 s$^{-1}$. For a sample of $L$ = 10 mm to compress 5 mm (strain 0.5) with such a strain rate, we would require a speed of 5 mm $\times$ 440 s$^{-1}$/ 0.5 strain = 4400 mm/s (22,000 faster than the current set up of 0.2 mm/s) in a time of 5 mm/4400 mm/ms = 0.00113 s. Such a test is impossible to achieve with a controlled acceleration and real compression force measurement as the machine should compensate for inertial forces to move the rigs. Therefore, impact tests are simple approaches for validating high strain rates. However, the strain rate is far from constant, as it starts and ends at zero, achieving a maximum strain rate around the region of interest of maximum acceleration.

## 5. Conclusions

The compression of cubes fabricated with a common desktop 3D printer using PLA, the most commonly used material, proved to be consistent in repetitions with a standard deviation in relation to force of just $\pm$3.6% in the worst case.

The scalability of force–displacement was confirmed, obtaining a very similar stress–strain curve and a design point level of 46% deformation with 75 MPa before increasing the load after leaving the plateau. The study of scalability proved that the orientation of fabrication had a negligible effect on stress–strain behavior. Scalability also provided a unique energy/mass curve with a design point of 21.6 kJ/kg or 25.92 kJ/cm$^3$.

Simulation and tests of compression compared well using a unique stress–stain curve even for points beyond the plateau region up to 50%. For the large cubes for which $L$ = 25 mm, the compression rig stopped at 50 kN. The simulations did not compare well in the elastic region as the purpose of this research was to focus on plastic deformation energy, which would imply a deceleration.

Simulations and tests of impacts using an energy design point within the plateau and scalability using the same unique stress–strain equation compared well for acceleration. They also compared well with theoretical values.

From a computational point of view, mass scaling to increase time step allowed for an accurate estimation of CPU time by using a constant time step. By using a large time step of 5 μs, the CPU time was reduced to around 31%. The use of a free time step provided very similar results, reducing the time step from the original 0.9 μs to 0.3 μs when the cube reached maximum compression of around 50%. With a forced time step, we avoided the risk of obtaining a very small time step due to a highly compressed element, which would slow down the entire simulation.

The stress–strain curve is easy to generate from experimental curves by using material 45 form. The introduction of plastic strain and stress for the common elastic-plastic material type 1 is more difficult as it requires the consideration of changes in cross-section during compression. Both materials are documented and compare well for the purpose of acceleration estimation.

The strain rates from the compression tests were as low as 0.02 s$^{-1}$, while during the impact tests, the strain rates could reach 440 s$^{-1}$, with a variation during the entire impact from the initial zero strain rate value. It is almost impossible to perform an experimental measurement of compression at such a constant strain rate as it would require a speed of

4.4 m/s during a displacement of just 5 mm. Therefore, the design of these impact tests allows the verification of high strain rates.

To summarize, herein, a protocol for performing impact tests on cubes at high strain rates is provided, allowing the prediction of acceleration through theory and simulation. Simulation provides a good approximation with a simple foam-type 45 material with a superimposed time step that reduces CPU time.

**Funding:** This research received no external funding.

**Data Availability Statement:** All simulations are available for download at https://meaagg.com/ESI/ESI_PAMCRASH.html (accessed on 20 February 2024).

**Acknowledgments:** The authors want to thank ESI for their continuous support regarding software implementation within IQS.

**Conflicts of Interest:** The authors declare no conflicts of interest.

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
