# Peer review of "High-Compression Crash Simulations and Tests of PLA Cubes Fabricated Using Additive Manufacturing FDM with a Scaling Strategy"

_computation, doi:10.3390/computation12030040_

Round 1

Reviewer 1 Report

Comments and Suggestions for Authors

The manuscript presents an in-depth analysis of compression tests conducted on PLA cubes generated by commonly used 3D printers. The study encompasses seven key objectives: ensuring repeatability, examining loading orientation, conducting comparisons through a single stress-strain curve, determining average values of acceleration, evaluating the time step, considering the materials model, and assessing the validity for a large strain model. The information disseminated in this manuscript holds significant value, and a comprehensive review of literature has been undertaken to establish the necessity of this research.

The findings indicate that, in general, desktop 3D printers can consistently produce high-quality PLA cubes. Even in the most adverse scenario, the force deviates by up to 4%. These conclusions are drawn from meticulous experimental observations. The reviewer expresses satisfaction and recommends acceptance of the manuscript, contingent upon addressing the following concerns:

1.       Please incorporate a scale bar into Figure 2. Additionally, designate the three figures used in this section as subsets, labeling them as a, b, and c.

2.       The manuscript requires a reorganization wherein all seven objectives should be explicitly stated in both the introduction and the experimental section. The reviewer was taken aback to find the objectives presented solely in the results section.

3.       Kindly provide a clear explanation for the notable disparity observed in the simulation results between Test_20 and SIM 20c.

4.       Figure 8: Please show the full graph.

5.       Table 4 should be included in the experimental section and not deferred to such a later stage in the discussion section in the manuscript.

Author Response

Thank you for your valuable comments. Please check file attached with my response,

Reviewer 2 Report

Comments and Suggestions for Authors

1) The described scope of the paper is misleading.  'Optimization of deformable parts...' is presented as one of the key research points of this study but the paper does not explore this direction at all. Only cube style samples are printed and the only distinguishing feature appears to be the size. No details are supplied on the build direction and the comment stating the 49% infill is used only for the L=10mm samples is confusing. So no (internal or external) geometry changes are explored which precludes the possibility of using the data for optimization. 

2) The compression tests are performed in a quasi static rate range which is far removed from impact situations. The strain energy is scaled to simulate higher rates. Viscoplastic effects can be significant with a decade or two increase in the strain rate, so I am averse to accept that the Instron compression data forms a representative stress-strain data set for use in the simulations. The authors are advised to explore high rate deformation data.

3) The inclined table test for determining friction may not be ideal. An upsetting experiment with cylindrical specimen form a standard protocol for determining the friction and would be more accurate for the actual test configuration.

4) How were other material constants needed for the finite element deformation determined? 

5) Figure captions need more details: is the data experimental or simulation? What are the deformation rates, need to clarify if stress is true or engineering in the axis labels.  

6) The entire discussion around equations 1-3 pertains to elastic deformation and this region is not considered in the net deformation energy. What is the point of including this in such detail?

Comments on the Quality of English Language

Shortcomings in the quality of the writing need to be addressed as the intended meaning is hard to understand in multiple places.  

Please find attached a marked copy of the manuscript.

Author Response

Thank you for your valuable comments. Please see my response in document attached.

Reviewer 3 Report

Comments and Suggestions for Authors

This work studies the compression and impact behavior of polymer PLA through experimentation and simulation. Some is the results is meaning for the field. However, the extensive amount of grammar issues in this manuscript is very distactive. And the results presentation needs improvements. Here are my questions and suggestions.

1: The authors emphasized that the sample was manufactured by 3D printing. In the introduction part, the authors also mentioned that 3D printed parts can be used for energy absorption applications due to its flexibility to create structures. However, I feel the results and discussion did not show the benefit or features of the 3D printed parts. In this paper, the authors used the 3D printing technique to print a solid cube. The authors tried to compare the difference between samples with different orientation. However, no obvious difference was found. What is the point of using 3D printing to create the cube instead of using a standard cast PLA block?

2: I suggest the authors check the scale, notes, line type of all the charts.

A: For figure 7, why there are multiple dash grey and dash yellow lines?

B: For figure 8, the lines are all overlapped together, which makes it hard to compare the experimental and simulation data for the same condition.

C: In figure 9, why there are vertical dashed blue and yellow lines at 0 s? Why was the mass of the simulation set to be different from the test one?

Comments on the Quality of English Language

The language needs extensive editing. There are many typos and grammar errors in the current manuscript. Some of these grammar issues made the content hard to understand.

Author Response

(The authors gave the same response as above.)

Round 2

Reviewer 1 Report

Comments and Suggestions for Authors

The authors made effort to improve the manuscript. It can be accepted now.

Author Response

Thank you for your valuable review. please find my detailed response attached.

Reviewer 2 Report

Comments and Suggestions for Authors

Several of the changes marked in the first version of the manuscript have not been addressed. 

Comments on the Quality of English Language

Needs to be proofread. 

Author Response

You may see the attachment

Reviewer 3 Report

Comments and Suggestions for Authors

I appreciate the effort put into the manuscript. However, there are still many typos grammar issues in the current manuscript.

For example, line 26, ‘nor’ should be ‘or’.

In line 29-31, the authors mentioned the deformation should be elastic to avoid a rebound. I believe elastic will cause rebound while plastic will avoid rebound.

In line 34-35, are tubes or egg boxed designed based on shell structures or not? In the current sentence, it suggests that tubes or egg boxes are examples of designs that are not based on shell structures.

In line 60-62, for several wall thicknesses should be with several wall thicknesses. And the whole sentence would be better if written in this way: They used tubes of inner diameter 51.04 mm with several wall thicknesses (1.6, 1.8, and 2 mm) and a length of 120 mm, applying compression at a rate of 2 mm/min (0.000277 s-1).

In line 70, strain rate should be added for speed rate of 2 mm/min.

In line 92, verification should be followed by of; if should be whether.

These are not all the grammar issues. I have observed several grammatical issues throughout the document that may impact the clarity and readability of the content. I recommend a thorough proofreading to address these concerns and ensure the manuscript meets the expected language standards for publication.

In addition,  In line 133-134, it was mentioned that the red arrow indicate the fabrication direction and black arrow indicate the loading direction. Why is the back arrow perpendicular to the loading diraction in figure 2? 

Comments on the Quality of English Language

As shown before.

Author Response

(The authors gave the same response as above.)

Round 3

Reviewer 2 Report

Comments and Suggestions for Authors

Concerns communicated in the previous review have been addressed.

Author Response

Dear reviewer 2. I can not find your comments for review 3 on Susy review system.I hope you were happy with my last review 2 report.

Reviewer 3 Report

Comments and Suggestions for Authors

The manuscript has been significantly improved, and I appreciate the efforts the authors have made. I only have one more suggestion. In line 490-491, the authors claimed that: 'Throughout this paper first, it was first demonstrated that additive manufacturing could produce repeatable force-displacement curves.' The authors can claim that their manufactured materials could produce repeatable force-displacement curves. However, this is not the first work in the world that has proven 3D printed structures can produce repeatable force-displacement curves. With this point being clarified, I agree for the paper to be published.

Author Response

Thank you for your comment. I just changed the sentence to clarify that we just wanted to demonstrate that we had repeatability for large compression rates which include filament detachment.